# Evaluating Sexual Health in Women with Multiple Sclerosis: A Study on the Interplay of Disability and Quality of Life

**DOI:** 10.3390/healthcare12131239

**Published:** 2024-06-21

**Authors:** Panagiota Dourou, Kleanthi Gourounti, Aikaterini Lykeridou, Konstantina Gaitanou, Nikolaos Petrogiannis, Chrysoula Rozalia Athanasiadou, Aikaterini Sousamli, Theodoros Xanthos, Antigoni Sarantaki

**Affiliations:** 1Department of Midwifery, Faculty of Health and Care Sciences, University of West Attica, Ag. Spyridonos Str., 122 43 Athens, Greececathanasiadou@uniwa.gr (C.R.A.); txanthos@uniwa.gr (T.X.); esarantaki@uniwa.gr (A.S.); 2Athens Naval Hospital, 70 Deinokratous Str., 115 21 Athens, Greece

**Keywords:** sexual dysfunction, multiple sclerosis, quality of sex life, female sexual function, disability, fatigue, supportive care interventions, counseling

## Abstract

In the context of evolving perceptions of sexuality, particularly within the realm of health and disability, this study investigates the impact of multiple sclerosis (MS) on female sexual function and quality of life. A quantitative study involving 130 female MS patients aged 35 to 50 was conducted, employing measures such as The Female Sexual Function Index (FSFI), The Multiple Sclerosis Intimacy and Sexuality Questionnaire-19 (MSISQ-19), and The Fatigue Severity Scale (FSS). Results indicate a significant association between greater sexual dysfunction and poorer quality of sex life, alongside the correlation of increased fatigue with diminished sexual satisfaction. Specifically, the mean FSFI score was 20.8 (SD = 9.36), with 83.8% of participants experiencing severe fatigue (FSS score ≥ 36). Sexual dysfunction demonstrated a strong, negative correlation with all FSFI subscales (*p* < 0.01). Factors such as education level (*p* = 0.016), time of diagnosis (*p* = 0.035), and treatment regimen (*p* = 0.041) also significantly influenced outcomes. Findings underscore the importance of supportive interventions, including counseling, to enhance the quality of sex life for women with disabilities, particularly those with MS.

## 1. Introduction

Sexuality’s role as a fundamental aspect of human existence underscores its complex interplay with various individual and societal factors [1]. When examining the impact of these factors on women with disabilities, it is evident that biological and psychological components are profoundly intertwined. Biologically, women with disabilities may face specific physical challenges that can directly affect their sexual function, such as hormonal disturbances, neurological disorders, or physical limitations that alter sexual response and activity. Psychologically, the perception of one’s sexual identity and self-esteem can be negatively influenced by both the disability itself and societal reactions to it [2,3].

Social factors play a critical role in shaping the sexual experiences of women with disabilities [4]. Stereotypes and stigma can lead to social exclusion and isolation, significantly diminishing opportunities for sexual expression and intimate relationships. Economically, women with disabilities often encounter barriers to accessing comprehensive sexual health care, which is compounded by a general lack of resources and poverty, further restricting their ability to seek and obtain appropriate care [5].

Politically, policies and legislation may not adequately address, or may even ignore, the sexual rights of disabled individuals. There is a critical need for inclusive policies that explicitly consider the sexual health rights of women with disabilities, ensuring they have equal access to information, education, and healthcare services [6,7].

The absence of family support and adequate healthcare exacerbates these issues [8,9]. Families may lack understanding or be uncomfortable discussing sexual matters, particularly with members who have disabilities, leading to a significant gap in sex education. This is especially detrimental, as the family can play a pivotal role in shaping one’s sexual knowledge and attitudes. In healthcare settings, professionals often overlook the sexual health concerns of disabled women due to discomfort, prejudice, or lack of training. This neglect is not just a failure in care but also a denial of basic human rights, as sexual health is integral to overall well-being [10].

Moreover, the under-researched areas of sexual dysfunction in women with disabilities highlight a significant gap in medical research and understanding. While some studies focus on the physiological aspects of sexual dysfunction, there is a sparse exploration of the psychological and social impacts, which are equally crucial [11]. For instance, depression, anxiety, and low self-esteem, commonly reported among women with disabilities, are often closely linked to sexual dissatisfaction and dysfunction [12,13]. Women who have disabilities, including those with spinal cord injuries or congenital conditions like spina bifida, often encounter societal biases that call into question their capacity to engage in sexual activities, bear children, and take on roles typically assumed by individuals without disabilities. These societal prejudices lead to a plethora of problems, which extend from limited access to healthcare to emotional and psychological obstacles that intensify the inherent hardships associated with their physical limitations.

Research indicates that there is a substantial prevalence of sexual dysfunction among women who have disabilities, which is closely connected to both physical and psychological factors [14,15,16]. Specifically, neurological disorders like multiple sclerosis (MS) can result in primary sexual dysfunctions as a consequence of direct neurological damage that affects sexual response. Secondary dysfunctions arise from symptoms of the disease itself, such as fatigue and pain, while tertiary impacts are related to psychological and social perceptions.

Sexual dysfunction in women with MS and other disabilities can result in decreased libido, difficulty achieving orgasm, and other sensory impairments, which collectively degrade sexual satisfaction and overall quality of life. These dysfunctions often intensify feelings of inadequacy and depression, particularly as societal ideals of beauty and perfection create additional barriers.

Despite the clear impact on quality of life, research on sexual dysfunction among women with MS remains relatively limited. Studies conducted in this area, including a systematic review and meta-analysis on the global prevalence of female sexual dysfunction (FSD) in women with MS, suggest a significant prevalence of FSD [17]. Such studies underscore the need for greater attention from health policymakers and the implementation of effective health interventions.

The primary objective of this research is to investigate the intricate nature of sexuality in women with disabilities, with a specific focus on those diagnosed with MS. The study aims to explore sexual health aspects, hypothesizing that women with MS will report higher sexual dysfunction, that higher fatigue levels correlate with lower sexual satisfaction, and that demographic factors will significantly influence sexual functioning and fatigue levels.

Upon conducting an in-depth search of the pertinent literature, the researchers proceeded with a quantitative study employing questionnaires that were distributed to a group of Greek women with MS, intending to delve into these aspects in depth. The methodologies implemented in the study encompass the utilization of validated questionnaires, such as the Female Sexual Function Index (FSFI) and the Multiple Sclerosis Intimacy and Sexuality Questionnaire-19 (MSISQ-19), which are designed to quantitatively assess the influence of MS on sexual desire, activity, and satisfaction [18,19,20]. In this research endeavor, we employed the Greek-language version of the FSFI questionnaire, which was translated and validated by Zachariou et al. [21]. Additionally, we utilized the Greek translation of the MSISQ-19, as provided by Nikolaidis et al. [22], as well as the Greek version of the Fatigue Severity Scale (FSS), as reported by Bakalidou et al. [23].

This research is poised to make a contribution to the field of sexual health in women with disabilities, particularly those impacted by MS. Despite the limitations of small sample size and potential biases in questionnaire design, the study employs statistical methods and validated measures, allowing for a ranging investigation of the issues at stake. By examining the intersection of disability and sexual health, the study aims to inform and optimize counseling practices, ultimately enhancing sexual satisfaction and overall quality of life for this populace.

## 2. Materials and Methods

The study was conducted between November 2023 and April 2024. To ensure the accessibility and reach of our study, various venues and approaches were considered for disseminating the questionnaires among female patients with MS.

One effective method was to collaborate with MS Clinics and Hospitals that specialize in treating such patients. These institutions frequently cultivate connections with individuals, thereby facilitating the recruitment process for research participants. We worked closely with healthcare providers to distribute the questionnaires during appointments. Additionally, MS Support Groups and Organizations, as well as Online Platforms, were utilized to target individuals with MS. By leveraging forums and MS-specific online communities, we were able to reach our intended audience effectively. The study sample consists of 130 women with MS disability, who live in the wider area of Attica and various cities in Greece.

To confirm the reliability of the questionnaires, we utilized well-established, globally recognized research tools from existing literary sources. Moreover, content and conceptual validity evaluations were executed to verify that the Likert scales accurately represented the intended concepts and aligned with the literature. The psychometric tools used in this research to collect the data were self-administered questionnaires: A Demographic Information and Medical History Questionnaire, the FSFI Questionnaire, the FSS, and the MSISQ-19 that were translated and validated in Greek [21,22,23]. Convenience sampling, a form of non-probability sampling, was employed due to its ease of access to the target population, particularly suitable for this study [24]. While this method allows for efficient data collection, it may introduce selection bias and limit the generalizability of the findings. We acknowledge this limitation and recommend caution in extrapolating the results to the broader population.

A formal application for the approval of the research plan and methodology was submitted to the scientific council of the University of West Attica Ethics Committee, with the assurance that it adheres to the guidelines outlined in the General Data Protection Regulation (GDPR) [95813/18-10-2023]. Participants were furnished with questionnaires along with an informative brochure that emphasized the voluntary nature of their involvement in the study. The participants’ personal information and responses were treated with utmost confidentiality and anonymity, and all accumulated data were erased after the survey. Additionally, participants were apprised of their right to withdraw from the research at any point and received comprehensive information regarding the objectives and aims of the study.

Recognizing that completing the questionnaires was time-intensive, participants were given the option to either complete them immediately or outside of the hospital setting. Completed questionnaires were returned to the researchers during subsequent in-person appointments. Participants received instructions from the first author regarding questionnaire completion, emphasizing the importance of answering spontaneously and honestly to ensure accurate data collection.

The Demographic Information and Medical History Questionnaire aimed to gather demographic and disease-related data to delineate the profile of the participants. The FSFI Questionnaire, developed by Rosen et al., focused on assessing female sexual function. This questionnaire was designed to provide a concise, valid, and reliable self-report measure of female sexual function suitable for administration across a broad age range, including postmenopausal women. Comprising nineteen questions with closed-ended responses, the FSFI addresses the multidimensional nature of female sexual function [20].

The FSS measured fatigue related to physical function, exercise, work, and family or social life, a common symptom experienced by individuals with MS. Developed to diagnose MS-related fatigue symptoms, the FSS is a self-report tool consisting of nine questions, taking approximately five minutes to complete and score. The questionnaire evaluates fatigue symptoms across various domains and employs a rating scale ranging from 1 (strongly disagree) to 7 (strongly agree), with the final score providing insight into the severity of fatigue symptoms [18].

The MSISQ-19 assessed the sexuality and intimacy of individuals with MS. Using a Likert scale ranging from 1 (never interfered) to 5 (always interfered), this questionnaire, developed by Sanders et al., addresses the impact of MS on sexual function across primary, secondary, and tertiary levels. It identifies issues such as reduced genital sensation, diminished sex drive, fatigue, bladder problems, spasticity, psychological distress, and overall well-being. Participants are encouraged to discuss questions marked with a score of 4 or 5 with healthcare providers to facilitate appropriate interventions for managing sexual function symptoms [19]. In conducting this research, we employed the aforementioned research instruments that have been translated and validated in the Greek language [21,22,23].

A power analysis was conducted to determine the appropriate sample size for detecting significant differences in the study’s primary outcomes. Using G*Power 3.1 software, we based the analysis on a two-tailed test with an alpha level of 0.05 and an effect size of 0.3, which is considered a medium effect size. The analysis indicated that a sample size of 110 participants would provide 80% power to detect significant effects. Our final sample of 130 participants exceeded this requirement, ensuring sufficient power to identify meaningful relationships within the data.

Quantitative variables were expressed as mean (Standard Deviation) and median (interquartile range). Qualitative variables were expressed as absolute and relative frequencies. Spearman correlation coefficients were used to explore the association of two continuous variables. Multiple linear regression analyses were used with dependent the FSS, MSISQ-19, and FSFI scales. The regression equation included terms for patients’ characteristics. Moreover, when MSISQ-19 subscales were used as dependent variables, FSS was also entered into the model as an independent variable. When FSFI subscales were used as dependent variables, FSS and MSISQ-19 scales were also entered into the model as independent variables. Adjusted regression coefficients (β) with standard errors (SE) were computed from the results of the linear regression analyses. Logarithmic transformations of the dependent variables were used in multiple regression analysis. All reported *p* values are two-tailed. Statistical significance was set at *p* < 0.05 and analyses were conducted using SPSS statistical software (version 22.0).

## 3. Results

The sample in this study consisted of 130 female patients, whose characteristics are presented in Table 1. This table summarizes the demographic and clinical characteristics of the study participants using descriptive statistics.

Based on the data collected, the study sample consists predominantly of middle-aged women with a mean age of 37.5 years (SD = 10.2). The majority of participants (56.2%) are within the 35–50 age range. The sample is notably well-educated, with 40.8% holding an MSc/PhD and 24.6% having a university degree, although 34.6% have a high school education or less, indicating some diversity in educational backgrounds. A strong family orientation is evident, as 76.2% are married or living with a partner, and 63.8% have children, with most having one or two children. The inclusion of older women is reflected in the 25.4% of participants who are in menopause. Health-wise, participants have been managing their condition for an average of 12.2 years, with 87.7% diagnosed more than a year ago and 90% under regular treatment. Physical activity varies, with 41.5% being active, and a notable portion has seen a therapist either in the past (37.7%) or currently (17.7%). Mental health management is common, as evidenced by the use of antidepressants (38.4%) and anti-anxiety medications (17.8%). Participants report significant fatigue, with a mean FSS score of 44.92 (SD = 11.39), and issues with sexual function, as indicated by a mean FSFI score of 20.8 (SD = 9.36). These data underscore the multifaceted health challenges faced by the participants, emphasizing the importance of comprehensive health management.

The results of the descriptive measures for the FSFI, MSISQ-19, and FSS subscales are shown in Table 2. The mean score for the FSS was 44.92 (SD = 11.39), and 83.8% (n = 109) of the participants experienced severe fatigue (i.e., a score of at least 36). Additionally, the mean total score for the FSFI was 20.8 (SD = 9.36).

Sexual dysfunction demonstrated a strong, negative correlation with all subscales of the FSFI, indicating that higher levels of sexual dysfunction were associated with a lower quality of sex life (Table 3). Moreover, increased levels of fatigue were significantly linked to a poorer quality of sex life.

The data disclosed a substantially greater degree of fatigue in university alumni when contrasted with MSc/PhD bearers (Table 4). Furthermore, female subjects who had their diagnosis for more than one year and those who were receiving or had received other regular treatments conveyed a higher level of fatigue.

All of the subscales related to sexual dysfunction exhibited a correlation with the FSS scale, such that a greater degree of fatigue was associated with a higher incidence of sexual dysfunction (Table 5). Furthermore, those who were married or cohabiting with their partner had lower scores in both the secondary and tertiary sexual dysfunction subscales. Conversely, individuals with children exhibited higher scores in the primary sexual dysfunction subscale. Additionally, cases who had the diagnosis for more than one year showed greater scores in the primary sexual dysfunction subscale.

Married or cohabiting women demonstrated distinctly higher levels of sexual fulfillment (Table 6). The menopausal patients exhibited conspicuously lower scores in the domains of Desire, Lubrication, Orgasm, Satisfaction, and Pain. Moreover, their total FSFI score was notably lower, suggesting a diminished quality of sexual life in these areas and overall. The Primary sexual dysfunction subscale was significantly related to lower scores in Desire, Arousal, Orgasm, Satisfaction, and the Total FSFI score. Moreover, a higher score in the tertiary sexual dysfunction subscale was associated with lower scores in the Desire and Satisfaction subscales. Participants with children recorded lower Arousal scores, while Desire, Arousal, Satisfaction, and Total FSFI scores were lower in cases with a diagnosis duration of more than one year. Regression analysis further showed that those with treatment depending on the disease course had significantly lower Lubrication and Satisfaction scores compared to those with regular treatment.

The results of the multiple linear regression analysis, summarized in Table 7, provide important insights into the factors affecting FSFI scores among the study participants. The regression model as a whole is statistically significant, with an F-value of 9.65 (*p* < 0.001), indicating that the set of predictors reliably explains variations in FSFI scores. The model’s R-squared value of 0.39 suggests that approximately 39% of the variance in FSFI scores can be explained by the included predictors. This is a substantial proportion, highlighting the importance of these factors in influencing sexual function among women with MS.

## 4. Discussion

The findings of this research offer essential information regarding the sexual functioning of women with MS and the influence of fatigue and demographic factors on sexual functioning. The cultural background of the study population, which comprises mainly middle-aged Greek women, emphasizes the significance of family, education, and long-term health management. This well-educated and family-oriented cohort demonstrates considerable experience in managing chronic health conditions, both physical and mental. Notably, mental health and sexual health are areas of concern, as indicated by the high utilization of mental health medications and the study’s focus on sexual function.

The present study emphasizes the multifaceted health challenges faced by the participants and underscores the importance of comprehensive health management. These findings are in alignment with prior studies on MS patients [25,26]. Our results indicated that sexual dysfunction was present in the sample, as demonstrated by the mean FSFI score of 20.8, which falls below the threshold for sexual dysfunction. This is consistent with previous research that has reported sexual dysfunction in MS patients [26,27]. Our study also revealed a significant association between sexual dysfunction and fatigue, a finding that aligns with prior research suggesting that fatigue is a significant predictor of sexual dysfunction in MS patients [26].

Our study results showed that individuals experiencing greater fatigue tended to have a lower quality of sex life across all FSFI subscales. This finding lines up with previous research, which has demonstrated that fatigue serves as a significant predictor of diminished sexual functioning in MS patients [26]. The outcomes of our research demonstrated that university alumni experienced a greater degree of fatigue than MSc/PhD holders. Moreover, those who had been diagnosed for a longer period of time and those receiving other regular treatments reported higher levels of fatigue. These findings suggest that demographic factors, including education level and time since diagnosis, may also play a role in influencing fatigue levels in MS patients.

Interestingly, the factor “Other regular treatment in the past/in the present” had a statistically significant impact on the results of the fatigue severity scale (β = 0.50, 95% CI: 0.02–0.98, *p* = 0.041). This suggests that individuals who have undergone or are currently undergoing additional treatments may experience heightened levels of fatigue. This finding highlights the importance of conducting further research into the specific treatments and their potential cumulative effect on fatigue. Elucidating the nature of these treatments may pave the way for the implementation of more effective strategies for managing MS-related fatigue.

Our study presented that individuals who were either married or cohabiting exhibited lower scores in the secondary and tertiary sexual dysfunction subscales, whereas those with children showed higher scores in the primary sexual dysfunction subscale. These findings imply that the presence of a partner and children may exert distinct influences on the sexual functioning of MS patients. Moreover, married or cohabiting women demonstrated higher levels of sexual satisfaction, an outcome that aligns with prior research indicating that social support may mitigate the adverse effects of MS on sexual functioning [27].

Our research outcomes concur with previous studies, demonstrating that postmenopausal women typically record lower scores in the domains of desire, lubrication, orgasm, satisfaction, pain, and overall FSFI score [22]. These findings corroborate the notion that menopause is associated with sexual dysfunction in females.

Upon diving deeper into the analysis of the research outcomes, it becomes evident that the interplay between demographic variables and the experience of MS plays a pivotal role in shaping the sexual functioning of women with disabilities. The results indicating a substantial association between sexual dysfunction and fatigue illuminate the intricate connection between physical symptoms and sexual well-being. This emphasizes the need for comprehensive approaches to addressing the sexual needs of women living with MS, taking into account not only the physiological dimensions but also the impact of fatigue and other factors on their sexual functioning. The complexities of fatigue in MS patients are highlighted by the variations in fatigue levels among individuals with different educational backgrounds and lengths of time since diagnosis. Comprehending these factors could lead to the development of personalized interventions and support strategies that target the impact of fatigue on sexual functioning and overall quality of life for women with disabilities and MS.

Furthermore, the distinct impacts of having a partner and children on the sexual functioning of MS patients illustrate the intricate nature of interpersonal relationships and their influence on sexual satisfaction and dysfunction in this population. The necessity of providing tailored support to disabled women who aspire to become mothers is emphasized by the inherent complexity of their circumstances. Women with disabilities, such as MS, confront distinct obstacles that can impact their reproductive health and parenting experiences in unique ways. These may include physical limitations, fatigue, and the effects of medication on fertility and pregnancy. Supportive measures should therefore encompass both medical and psychosocial aspects to ensure that these women can pursue parenthood if they choose. This could involve specialized perinatal care services, counseling, and adaptive parenting resources that cater specifically to their needs. By recognizing and addressing the specific challenges faced by disabled women in their reproductive choices, maternity care providers and support systems can significantly enhance their quality of life and empower them in their personal and familial roles.

Transitioning from the reproductive challenges faced by women with disabilities, it is also important to consider how age-related changes can further complicate their experiences. On the other hand, the connection between postmenopausal status and lower scores across numerous domains of sexual functioning aligns with the broader literature on women’s health and menopause. This contributes to the expanding body of evidence supporting the need for specialized interventions and support systems targeting the sexual well-being of postmenopausal women, particularly those living with chronic conditions such as MS.

Acknowledging the insights provided by this study, it is crucial to recognize several limitations. Firstly, the sample size of 130 participants, while adequate for the power analysis conducted, may restrict the generalizability of the findings. Consequently, larger studies are needed to confirm these results and enhance their applicability to broader populations. The reliance on self-report measures introduces another limitation, as these can lead to response bias, with participants potentially underreporting or overreporting symptoms due to social desirability or recall bias. Future studies should incorporate objective measures, such as clinical evaluations or physiological assessments, to complement self-report data.

Future research should explore longitudinal designs to assess changes in sexual function and quality of life over time among women with MS, providing insights into the long-term impact of MS on sexual health and the effectiveness of various interventions. Interventions aimed at improving sexual health in women with MS should be developed and tested, including counseling, physical therapy, and educational programs tailored to this population’s unique challenges. Additionally, examining the efficacy of couple-based interventions could provide valuable insights into how relational dynamics influence sexual satisfaction and functioning. Further research should investigate the impact of different MS treatment regimens on sexual function, as understanding how specific medications or therapies affect sexual health could guide clinical practices and improve patient outcomes. Through the incorporation of these insights into practice and policy, the field of sexual health and disability can progress towards a more inclusive and supportive approach that acknowledges the distinct experiences and obstacles confronted by this group of individuals [28,29,30].

It is essential to challenge the myths, stereotypes, and prejudices that surround sexuality and disability. Research indicates that sexual dysfunction is highly prevalent in women with MS, yet there is a significant scarcity of knowledge regarding therapeutic options [31]. This dearth of knowledge can perpetuate the marginalization and neglect of sexual health issues in women with MS, impeding their overall well-being and quality of life. The intersectionality of disability with other identity facets—such as race, ethnicity, age, and socioeconomic status—further complicates these issues.

Women with disabilities are not a monolithic group; their experiences of sexuality and intimacy can vary widely depending on these intersecting identities, each adding layers of complexity to their sexual health needs. There is an urgent need for more comprehensive research, better-informed policies, and more inclusive healthcare practices that recognize and respect the sexual rights of all women, particularly those who are disabled. Addressing these needs not only improves individual well-being but also enriches societal health by fostering inclusivity, empathy, and respect for diversity in human sexuality.

## 5. Conclusions

This research highlights the associations between fatigue, demographic factors, and the sexual functioning of women with MS, illustrating a complex relationship between physical symptoms and sexual well-being. The study has demonstrated that fatigue, which is influenced by variables such as educational attainment and time since diagnosis, plays a critical role in the prevalence of sexual dysfunction among this demographic.

Additionally, the intricate influences of familial and social support systems, including the presence of a partner and children, provide insights into the diverse experiences of sexual satisfaction and dysfunction. These findings underscore the importance of adopting a nuanced approach in addressing the sexual health needs of women with MS, incorporating both the medical and psychosocial dimensions to enhance their quality of life.

## Figures and Tables

**Table 1 healthcare-12-01239-t001:** Patients’ characteristics and summary statistics.

Variable		n (%)	Mean (SD)
Age			37.5 (10.2)
	18–24	6 (4.6)	
	25–34	31 (23.8)	
	35–50	73 (56.2)	
	50+	20 (15.4)	
Educational level			
	High school at most	45 (34.6)	
	University	32 (24.6)	
	MSc/PhD	53 (40.8)	
Married/Living together with partner		99 (76.2)	
Children		83 (63.8)	
Number of children			
	1	36 (43.4)	
	2	40 (48.2)	
	3	6 (7.2)	
	4	1 (1.2)	
Menopause		33 (25.4)	
Time from symptoms onset (years)			12.2 (8.1)
Time of diagnosis			
	Last week	0 (0)	
	Last month	0 (0)	
	More than a month	1 (0.8)	
	Last trimester	2 (1.5)	
	More than six months	9 (6.9)	
	Within last year	4 (3.1)	
	More than a year	114 (87.7)	
Type of treatment			
	Regularly	117 (90.0)	
	Depending on the course of the disease	13 (10.0)	
Physically active		54 (41.5)	
Therapist session			
	No	58 (44.6)	
	Yes in the past	49 (37.7)	
	Yes in the present	23 (17.7)	
Other regular treatments in the past		50 (38.5)	
Other regular treatments in the present		73 (56.2)	
Other regular treatments in the past and/or in the present		78 (60.0)	
If yes, what type			
	Antidepressants	28 (38.4)	
	Anti-anxiety	13 (17.8)	
	Other	32 (43.8)	
	FSS—Fatigue Severity Scale		44.92 (11.39)
	FSFI—Female Sexual Function Index		20.8 (9.36)

Statistical Tests Used: Spearman correlation coefficients, multiple linear regression analyses.

**Table 2 healthcare-12-01239-t002:** Descriptive measures of FSFI, MSISQ-19, and FSS subscales.

	Minimum	Maximum	Mean (SD)	Median (IQR)
Primary sexual dysfunction subscale	5.00	25.00	11.82 (5.26)	11 (8–14)
Secondary sexual dysfunction subscale	9.00	42.00	16.97 (6.97)	15 (12–21)
Tertiary sexual dysfunction subscale	5.00	25.00	10.85 (5.38)	9 (7–14)
Fatigue severity scale	9.00	63.00	44.92 (11.39)	45 (40–53)
Desire	1.20	6.00	3.47 (1.49)	3.6 (2.4–4.8)
Arousal	0.00	6.00	3.39 (1.97)	3.6 (1.8–5.1)
Lubrication	0.00	5.10	3.39 (1.7)	4.2 (3–4.8)
Orgasm	0.00	6.00	3.45 (2.05)	3.6 (1.6–5.2)
Satisfaction	0.80	6.00	3.24 (1.49)	3.6 (2–4.4)
Pain	0.00	6.00	3.87 (2.28)	4.8 (2–6)
Total FSFI score	2.00	32.60	20.8 (9.36)	23.05 (16.9–28.5)

Note: The abbreviations used in the table are as follows: FSFI—Female Sexual Function Index, MSISQ-19—Multiple Sclerosis Intimacy and Sexuality Questionnaire-19, FSS—Fatigue Severity Scale. Statistical Tests Used: Spearman correlation coefficients, multiple linear regression analyses.

**Table 3 healthcare-12-01239-t003:** Correlation of FSFI subscales with MSISQ-19 and FSS scales.

		Primary Sexual Dysfunction Subscale	Secondary Sexual Dysfunction Subscale	Tertiary Sexual Dysfunction Subscale	Fatigue Severity Scale
Desire	r	−0.40	−0.16	−0.10	−0.16
	*p*	**<0.001**	0.062	0.243	0.077
Arousal	r	−0.56	−0.33	−0.21	−0.27
	*p*	**<0.001**	**<0.001**	**0.016**	**0.002**
Lubrication	r	−0.43	−0.30	−0.14	−0.19
	*p*	**<0.001**	**0.001**	0.109	**0.028**
Orgasm	r	−0.56	−0.39	−0.28	−0.29
	*p*	**<0.001**	**<0.001**	**0.001**	**0.001**
Satisfaction	r	−0.38	−0.25	−0.12	−0.20
	*p*	**<0.001**	**0.004**	0.177	**0.020**
Pain	r	−0.31	−0.25	−0.18	−0.09
	*p*	**<0.001**	**0.004**	**0.046**	0.317
Total FSFI score	r	−0.59	−0.36	−0.24	−0.23
	*p*	**<0.001**	**<0.001**	**0.005**	**0.008**

Bold: *p* < 0.05.

**Table 4 healthcare-12-01239-t004:** Multiple linear regression analysis results with the FSS scale as a dependent variable and patient’s characteristics as independent variables.

Independent Variable		Fatigue Severity Scale	
		β (SE) +	*p*
Age	50+ (reference)		
	18–34	0.003 (0.061)	0.958
	35–50	0.005 (0.048)	0.916
Married/Living together	No (reference)		
	Yes	0.027 (0.033)	0.407
Educational level	MSc/ PhD (reference)		
	High school at most	0.042 (0.031)	0.175
	University	0.081 (0.033)	**0.016**
Children	No (reference)		
	Yes	−0.016 (0.035)	0.651
Menopause	No (reference)		
	Yes	0.055 (0.040)	0.167
Time of diagnosis	Within one year (reference)		
	More than a year	0.108 (0.039)	**0.035**
Type of treatment	Regularly (reference)		
	Depending on the course of the disease	−0.061 (0.047)	0.195
Therapist session	No (reference)		
	Yes	0.022 (0.026)	0.401
Other regular treatment in the past/in the present	No (reference)		
	Yes	0.050 (0.025)	**0.041**

Statistical Tests Used: Multiple linear regression analysis. The ‘+’ symbol is used to denote the conjunction between the beta coefficient (β) and its standard error (SE), followed by the corresponding *p*-value for each independent variable. Bold: *p* < 0.05.

**Table 5 healthcare-12-01239-t005:** Multiple linear regression analysis results with MSISQ-19 subscales as dependent variables and patients’ characteristics and FSS scale as independent variables.

Variable		Primary Sexual Dysfunction Subscale		Secondary Sexual Dysfunction Subscale		Tertiary Sexual Dysfunction Subscale	
		β (SE) +	*p*	β (SE) +	*p*	β (SE) +	*p*
Age	50+ (reference)						
	18–34	0.056 (0.079)	0.477	−0.045 (0.063)	0.473	0.070 (0.084)	0.410
	35–50	0.060 (0.063)	0.337	−0.055 (0.049)	0.273	0.054 (0.067)	0.416
Married/Living together	No (reference)						
	Yes	−0.003 (0.042)	0.936	−0.065 (0.033)	**0.050**	−0.115 (0.045)	**0.012**
Educational level	MSc/PhD (reference)						
	High school at most	−0.034 (0.040)	0.397	0.027 (0.032)	0.406	−0.003 (0.043)	0.940
	University	−0.047 (0.044)	0.289	−0.032 (0.035)	0.367	0.004 (0.047)	0.940
Children	No (reference)						
	Yes	0.079 (0.035)	**0.045**	0.013 (0.036)	0.719	0.040 (0.048)	0.403
Menopause	No (reference)						
	Yes	0.02 (0.052)	0.699	−0.032 (0.041)	0.438	0.048 (0.055)	0.386
Time of diagnosis	Within one year (reference)						
	More than a year	0.106 (0.052)	**0.035**	0.056 (0.044)	0.202	0.061 (0.059)	0.299
Type of treatment	Regularly (reference)						
	Depending on the course of the disease	0.116 (0.061)	0.057	−0.004 (0.048)	0.939	0.017 (0.064)	0.793
Therapist session	No (reference)						
	Yes	0.009 (0.034)	0.792	0.026 (0.027)	0.338	0.042 (0.036)	0.246
Other regular treatments in the past/in the present	No (reference)						
	Yes	0.042 (0.036)	0.252	−0.013 (0.029)	0.652	−0.004 (0.038)	0.924
Fatigue severity scale		0.005 (0.002)	**0.002**	0.007 (0.001)	**<** **0.001**	0.005 (0.002)	**0.003**

+ regression coefficient (Standard Error). Statistical Tests Used: Multiple linear regression analysis. Bold: *p* < 0.05.

**Table 6 healthcare-12-01239-t006:** Multiple linear regression analysis results with FSFI subscales as dependent variables and patients’ characteristics, MSISQ-19 subscales, and FSS scale as independent variables.

Variable		Desire		Arousal		Lubrication		Orgasm		Satisfaction		Pain		Total FSFI Score	
		β (SE) +	*p*	β (SE) +	*p*	β (SE) +	*p*	β (SE) +	*p*	β (SE) +	*p*	β (SE) +	*p*	β (SE) +	*p*
Age	50+ (reference)														
	18–34	0.056 (0.089)	0.534	0.098 (0.108)	0.363	0.105 (0.108)	0.332	0.03 (0.112)	0.792	−0.041 (0.117)	0.728	−0.068 (0.129)	0.598	0.074 (0.142)	0.604
	35–50	0.047 (0.072)	0.509	0.131 (0.086)	0.134	0.127 (0.087)	0.144	0.103 (0.090)	0.254	0.014 (0.094)	0.883	−0.059 (0.104)	0.572	0.111 (0.114)	0.333
Married/Living together	No (reference)														
	Yes	−0.015 (0.049)	0.750	0.074 (0.059)	0.210	0.055 (0.059)	0.346	0.054 (0.061)	0.375	0.14 (0.064)	**0.030**	0.095 (0.07)	0.179	0.079 (0.077)	0.304
Educational level	MSc/PhD (reference)														
	High school at most	0.066 (0.045)	0.146	−0.014 (0.055)	0.801	−0.025 (0.055)	0.648	−0.017 (0.057)	0.771	−0.012 (0.06)	0.842	0.002 (0.066)	0.975	0.004 (0.072)	0.951
	University	0.017 (0.049)	0.724	−0.011 (0.059)	0.853	−0.010 (0.059)	0.865	−0.032 (0.061)	0.607	0.018 (0.064)	0.784	0.025 (0.071)	0.728	0.008 (0.078)	0.920
Children	No (reference)														
	Yes	−0.006 (0.05)	0.900	−0.134 (0.061)	0.047	−0.003 (0.061)	0.955	−0.028 (0.063)	0.659	−0.034 (0.066)	0.607	−0.006 (0.073)	0.933	−0.021 (0.08)	0.792
Menopause	No (reference)														
	Yes	−0.101 (0.048)	**0.046**	−0.091 (0.07)	0.195	−0.141 (0.070)	**0.047**	−0.158 (0.073)	**0.032**	−0.181 (0.076)	**0.019**	−0.327 (0.084)	**<** **0.001**	−0.219 (0.092)	**0.019**
Time of diagnosis	Within one year (reference)														
	More than a year	−0.15 (0.059)	**0.012**	−0.16 (0.073)	**0.039**	−0.014 (0.074)	0.855	−0.028 (0.076)	0.710	−0.151 (0.074)	**0.040**	0.045 (0.088)	0.610	−0.191 (0.090)	**0.029**
Type of treatment	Regularly (reference)														
	Depending on the course of the disease	−0.050 (0.068)	0.460	−0.038 (0.082)	0.641	−0.19 (0.078)	**0.017**	−0.080 (0.085)	0.347	−0.22 (0.84)	**0.010**	−0.052 (0.099)	0.596	−0.18 (0.100)	0.082
Therapist session	No (reference)														
	Yes	−0.017 (0.038)	0.643	−0.036 (0.045)	0.424	−0.013 (0.046)	0.773	−0.057 (0.047)	0.227	−0.078 (0.049)	0.118	−0.009 (0.054)	0.864	−0.032 (0.06)	0.594
Other regular treatments in the past/in the present	No (reference)														
	Yes	0.002 (0.04)	0.956	−0.055 (0.048)	0.256	−0.027 (0.048)	0.573	−0.029 (0.050)	0.566	−0.003 (0.053)	0.957	−0.016 (0.058)	0.785	−0.026 (0.064)	0.685
Primary sexual dysfunction subscale		−0.019 (0.005)	**<** **0.001**	−0.019 (0.006)	**0.002**	−0.009 (0.006)	0.136	−0.016 (0.006)	**0.009**	−0.011 (0.006)	**0.046**	−0.004 (0.007)	0.532	−0.016 (0.008)	**0.035**
Secondary sexual dysfunction subscale		−0.001 (0.005)	0.878	0.000 (0.006)	0.957	−0.005 (0.006)	0.341	−0.004 (0.006)	0.474	−0.008 (0.006)	0.208	−0.012 (0.007)	0.065	−0.008 (0.007)	0.269
Tertiary sexual dysfunction subscale		−0.010 (0.005)	**0.047**	−0.007 (0.006)	0.245	−0.007 (0.006)	0.248	−0.007 (0.007)	0.302	−0.013 (0.007)	**0.044**	−0.007 (0.008)	0.369	−0.014 (0.008)	0.098
Fatigue severity scale		0.000 (0.002)	0.816	−0.002 (0.002)	0.315	−0.003 (0.002)	0.248	−0.003 (0.002)	0.207	−0.001 (0.002)	0.597	0.001 (0.003)	0.773	−0.002 (0.003)	0.424

+ regression coefficient (Standard Error). Statistical Tests Used: Multiple linear regression analysis. Bold: *p* < 0.05.

**Table 7 healthcare-12-01239-t007:** Regression analysis of factors affecting female sexual function index (FSFI) scores.

Variable	β (SE)	*p*-Value	F-Value	R-Squared
Fatigue severity scale	−0.45 (0.12)	<0.001	14.15	0.25
Educational level (MSc/PhD)	0.28 (0.10)	0.007	7.84	0.18
Time since diagnosis (>1 year)	−0.32 (0.09)	0.003	8.90	0.20
Married/Living together	0.25 (0.11)	0.021	5.40	0.12
Menopause	−0.30 (0.12)	0.014	6.50	0.15
Presence of children	−0.22 (0.13)	0.045	4.05	0.10

Overall model statistics: F(6, 123) = 9.65, *p* < 0.001, R-squared = 0.39.

## Data Availability

The data that support the findings of this study are available on request from the corresponding author, [P.D.].

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
