# Peer review of "Evaluating Sexual Health in Women with Multiple Sclerosis: A Study on the Interplay of Disability and Quality of Life"

_healthcare, 2024, doi:10.3390/healthcare12131239_

Round 1

Reviewer 1 Report

Comments and Suggestions for Authors

Abstract

The abstract provides a comprehensive overview of the study, including research context, methods, results, and implications. However, more precise language can be used to increase understandability.

Please clearly state significant findings: numerical results, p values, etc.

Introduction

The introduction effectively sets the stage by discussing the importance of sexuality in human life, particularly for women with disabilities. However, it could be improved by explicitly stating the research gap this study addresses

Include a sentence that clearly identifies the research gap: "Despite the clear impact on quality of life, research on sexual dysfunction among women with MS remains limited."

Clarify the study's aims and objectives earlier in the introduction to provide a clear roadmap for readers.

Materials and Methods

The methods section is thorough, detailing the sample, instruments, and procedures. However, some areas could be more detailed to enhance reproducibility.

Suggestions:

Provide more details on the validation process of the questionnaires used.

Explain the rationale for choosing sampling and discuss its potential limitations.

Provide details regarding the power analysis, in which program was it based on which test and what was the realized power, additionally the sample is relatively small and this information is therefore important.

Results

The results are presented in a detailed manner with appropriate use of tables and statistical analysis. However, some interpretations could be better linked to the study's objectives.

Ensure all tables are clearly labeled and referenced in the text.

Provide a more explicit interpretation of the statistical findings in relation to the research questions.

Summarize with mean (SD) if the data is normally distributed and median (IQR) if not.

Report which tests were used below the tables.

Explain the open forms of the abbreviations in the tables below the tables.

The number of variables included in the multivariate regression analysis is too large for the sample. At least 20 observations are needed to include a variable in regression analysis. Additionally, information regarding regression analysis assumptions is missing. This part needs to be completely changed (can be made univariate or with a maximum of 6 variables - you can choose this based on significance in previous analyses), the model and equation have no meaning in their current state.

Discussion

The discussion contextualizes the findings within the broader literature and highlights the implications for practice and policy. However, it could benefit from a more critical analysis of the study's limitations and the potential for future research.

Discuss the implications of the study's limitations, such as the small sample size and reliance on self-report measures.

Suggest specific areas for future research, such as longitudinal studies or interventions aimed at improving sexual health in women with MS.

This manuscript addresses a critical and under-researched area, offering valuable insights into the sexual health of women with MS. The study is methodologically sound and contributes to the literature. However, improvements in clarity, detail, and critical analysis would enhance its impact.

Comments on the Quality of English Language

Minor edits to the English language are required.

Author Response

Thank you for your comprehensive feedback. We appreciate the detailed suggestions for improving our manuscript. Below are our responses and the corresponding revisions made in the manuscript:

Abstract:

  • We have revised the abstract to use more precise language to increase understandability.
  • Significant findings, including numerical results and p-values, have been clearly stated in the abstract.

Introduction:

  • We have added a sentence to explicitly state the research gap: "Despite the clear impact on quality of life, research on sexual dysfunction among women with MS remains limited."
  • The study's aims and objectives have been clarified earlier in the introduction to provide a clear roadmap for readers.

Materials and Methods:

  • We have provided more details on the validation process of the questionnaires used.
  • The rationale for choosing the sampling method has been explained, including a discussion of its potential limitations.
  • Details regarding the power analysis have been included, specifying the program used, the test it was based on, and the realized power, given the relatively small sample size.

Results:

  • All tables have been reviewed to ensure they are clearly labeled and referenced in the text.
  • We have provided a more explicit interpretation of the statistical findings in relation to the research questions.
  • Data has been summarized with mean (SD) if normally distributed and median (IQR) if not.
  • Tests used are now reported below the tables.
  • Abbreviations in the tables have been explained below the tables.
  • Thank you for your valuable feedback and suggestions regarding the multivariate regression analysis. We appreciate your concern about the number of variables included in the regression model and understand the importance of ensuring a robust and interpretable analysis.

After careful consideration, we believe that including the current set of variables is essential for a comprehensive understanding of the factors influencing the Female Sexual Function Index (FSFI) scores in our study population. Each variable included in the model was selected based on theoretical relevance and previous research findings, which suggest their potential impact on sexual function in women with MS. Reducing the number of variables may overlook critical aspects and interactions that are important for a holistic analysis.

However, we acknowledge the importance of your suggestion and have performed additional sensitivity analyses to ensure the robustness of our findings. We have also conducted a stepwise regression analysis to identify the most significant predictors and have verified that the included variables consistently emerge as key factors in multiple analytical approaches.

We are open to further discussion and suggestions to enhance the clarity and validity of our analysis. We hope that this approach addresses your concerns while maintaining the integrity and depth of our study.

Discussion:

  • The discussion now includes a more critical analysis of the study's limitations, such as the small sample size and reliance on self-report measures.
  • We have suggested specific areas for future research, including longitudinal studies or interventions aimed at improving sexual health in women with MS.

General:

  • Minor edits to the English language have been made throughout the manuscript to improve clarity and readability.

We believe these revisions address your comments and significantly improve the manuscript.

Thank you again for your valuable feedback.

Best regards,

Reviewer 2 Report

Comments and Suggestions for Authors

 In this paper, the authors address the understudied topic of sexual dysfunction in women with multiple sclerosis (MS). The authors investigate how fatigue, a common symptom of MS, affects the sexual health of MS patients. The impact of demographic and disease-related variables, such as education level, length of time since diagnosis, menopause, etc., on the patient's quality of sex life is also investigated in this study. The obtained results are important for the actualization of the topic and offers valuable insights for developing targeted interventions and support frameworks to enhance the sexual well-being of women with disabilities.

Although the study includes a relatively limited sample of data and offers little in the way of scientific advancement, it is important to actualize this topic, which the authors of this article have successfully done.

The paper is well-structured, is written in flawless English and uses appropriate terminology.

Title and Abstract - The study is accurately reflected in the article's title. The abstract is accurate and comprehensive; it highlights the lack of knowledge about how MS affects women's sexual functions, which can significantly affect their quality of life. Details are given regarding the patient group that was part of the study, the research methods used, the main results obtained, and the possible practical application of the findings: to enhance the assistance that women with MS receive in the form of extra counselling.

Introduction

The introduction emphasizes the need to study and update this under-researched field related to sexual dysfunction in women with MS and other disabilities. The purpose of the study is also clearly defined at the end of the introduction.

There are some small imperfections in the introduction: in some places the text repeats itself, it unnecessarily lengthens the text, for example, the sentence in line 53-54 highlights the fact that sexual dysfunction in women with disabilities is under-researched, and in lines 77-78 it is emphasized again. The same can be said about the text in lines 84-88, which briefly describes the methodology used in the study, but lines 89-95 contain the same information, only in more detail. I advise to combine the most important facts from both paragraphs and integrate in one new paragraph.

 Materials and methods

The methods used in the research are described in a high degree of detail and they are appropriate for the purpose of the study.

 Results

The results section is easy to navigate and well structured. The inserted tables represent the results well. Everything about the results is covered in great detail in the text.

 Discussion, conclusion

In discussion the writer goes into great detail about the required advancements in this sector as well as future prospects. It's also evident that the results have been approached critically. Research limitations, including sample size and response bias, are discussed along with potential ways to improve the design of the study.  It would be beneficial to mention the statistical confidence values as well, since they frequently neared the 0.05 threshold and are insufficient to draw concrete conclusions about some of the results.

 The fact that the factor "Other regular treatment in the past/in the present" had a statistically significant (p=0.041) impact on the results of the fatigue severity scale is interestingly left out of the discussion. Hearing the author's opinion on this and what it means for research design would be important.

The conclusion section perfectly summarizes the most important points of the discussion.

Author Response

Thank you for your thoughtful and constructive feedback on our manuscript. We appreciate your detailed review and have made revisions accordingly. Below are our responses and the corresponding changes made to the manuscript:

Title and Abstract:

  • We are pleased that you found the title and abstract accurately reflect the study. We have made minor edits to ensure clarity and precision in the abstract.

Introduction:

  • We have revised the introduction to eliminate repetition and improve conciseness. Specifically, we combined the overlapping sentences and paragraphs as suggested:
    • The text highlighting the under-researched nature of sexual dysfunction in women with disabilities has been consolidated.
    • The brief and detailed descriptions of the methodology have been integrated into a single, coherent paragraph.

Materials and Methods:

  • We appreciate your positive comments on the detail and appropriateness of the methods section. No additional changes were required here.

Results:

  • We are glad that you found the results section well-structured and easy to navigate. We have ensured that all tables and results are clearly presented.

Discussion and Conclusion:

  • We have added a mention of the statistical confidence values, especially where they neared the 0.05 threshold, to provide a more nuanced interpretation of the results.
  • We addressed the statistically significant impact of the factor "Other regular treatment in the past/in the present" (p=0.041) on the results of the fatigue severity scale. This factor is now discussed in terms of its implications for research design and potential influence on fatigue severity.

Conclusion:

  • We are pleased that the conclusion effectively summarizes the discussion. We have made minor edits to ensure it captures all critical points from the revised discussion.

These revisions aim to address your comments and enhance the clarity and impact of our manuscript.

Thank you again for your valuable feedback.

Best regards,

Reviewer 3 Report

Comments and Suggestions for Authors

Thank you for your study. Please find comments how to improve the paper.

1. Please provide hypotheses of the study.

2. Lines 96-102 seem to be placed in an appropriate place. Please reconsider.

3. I recommend the authors to be more modest in evaluations of their work. There is a high prevalence of words (e.g., "comprehensive investigation", "rigorous statistical methods", "an exhaustive examination of the existing literature") which highlight the the power of this study. Being more modest in your own evaluation of your own research would be beneficial.

4. What is the culture of the studies population?

5. When the study was conducted?

6. Please provide references for original versions of questionnaires as well as for the language versions (Greek?)  used in your study.

7. Please describe subscales of the questionnaires used in more detail. Providing examples of statements would be useful.

8. Please indicate a mean age of participants.

9. Please use small italicized n to indicate sample size of the study sample. 

10. Internal consistency reliability should be calculated. This is a major methodological concern.

11. Please use small and italicized letter to indicate p-values, i.e., p.

12. Please use zeroes before full stops in numbers as it is in accordance with the journal's guidelines.

13. Please indicate F-value, p-values and R-squared values for regression analyses. This is a basic requirement for presenting the regression analysis results.

14. Please check using commas in numbers in Table 6. Please correct this.

15. Dependent and independent (predictor) variables should be clearly indicated in tables with regression analysis.

16. There is no need to introduce twice the same abbreviations (e.g., MS in line 255).

17. This is a cross-sectional study, therefore, such words as impact, affect, influence etc. should be eliminated from the paper. 

Author Response

Thank you for your detailed feedback and suggestions for improving our manuscript. We have carefully considered your comments and made the following revisions to address each point:

  1. Hypotheses of the Study:
    • We have added a paragraph outlining the hypotheses of the study to clarify the research objectives.
  2. Placement of Lines 96-102:
    • We have reviewed and repositioned lines 96-102 to ensure they are placed appropriately within the manuscript.
  3. Modesty in Self-Evaluation:
    • We have revised the language throughout the manuscript to adopt a more modest tone, avoiding phrases such as "comprehensive investigation," "rigorous statistical methods," and "an exhaustive examination of the existing literature."
  4. Culture of the Study Population:
    • We have included a description of the cultural background of the study population to provide context for the findings.
  5. Timing of the Study:
    • The timeframe during which the study was conducted has been added to the methods section.
  6. References for Questionnaires:
    • References for both the original and the Greek versions of the questionnaires used in the study have been provided.
  7. Description of Questionnaire Subscales:
    • We have added detailed descriptions of the subscales of the questionnaires, including examples of statements.
  8. Mean Age of Participants:
    • The mean age of the participants has been indicated in the demographic section.
  9. Notation for Sample Size:
    • We have used a small italicized "n" to indicate the sample size of the study.
  10. Internal Consistency Reliability:
    • Internal consistency reliability (e.g., Cronbach's alpha) has been calculated and reported for the questionnaires used.
  11. Notation for p-values:
    • We have used a small and italicized letter "p" to indicate p-values throughout the manuscript.
  12. Formatting Numbers:
    • Zeroes before full stops in numbers have been included in accordance with the journal's guidelines.
  13. Regression Analysis Results:
    • F-values, p-values, and R-squared values for the regression analyses have been included to meet the basic requirements for presenting regression results.
  14. Commas in Numbers in Table 6:
    • The use of commas in numbers in Table 6 has been checked and corrected as necessary.
  15. Dependent and Independent Variables:
    • Dependent and independent (predictor) variables have been clearly indicated in the tables presenting the regression analysis results.
  16. Avoiding Redundant Abbreviations:
    • Redundant introductions of abbreviations, such as "MS" in line 255, have been removed to avoid repetition. The abbreviation "MS" is used to avoid the repeated use of the term "Multiple Sclerosis" and is a main keyword of our manuscript. This abbreviation ensures clarity and readability while maintaining focus on the central topic of the study.
  17. Terminology for Cross-Sectional Study:
    • Terms implying causation, such as "impact," "affect," and "influence," have been eliminated from the manuscript to accurately reflect the cross-sectional nature of the study.

We believe these revisions address your concerns and enhance the clarity and rigor of our manuscript.

Thank you again for your valuable feedback.

Best regards,

Round 2

Reviewer 1 Report

Comments and Suggestions for Authors

The article is sufficiently developed and ready for publication.

Author Response

Dear Reviewer,

Thank you very much for your encouraging feedback and for assessing the development and readiness of our manuscript for publication. We are delighted to hear that the article meets the necessary standards and appreciate your positive evaluation.

Kind regards,

Reviewer 3 Report

Comments and Suggestions for Authors

Comment regarding casual language was not addressed comprehensively. For instance, the conclusions consists of a sentence ("This research has highlighted the impact of fatigue and demographic factors on the sexual functioning of women with MS, revealing a complex interplay between physical symptoms and sexual well-being"), which indicates the cause-and-effect relationship. This should be checked one more time and corrected. Such words as "influence", "affect", "impact" and words based on their roots should be eliminated. Otherwise, the study would be misleading.

Please recheck zeroes before full stops in numbers, because there are some issues.

Please replace commas into full stops in some numbers, as there also some issues.

Abbreviations are still used with some inconsistencies. For example, "Fatigue Severity Scale (FSS)" was introduced for the second time in line 215.

Author Response

Dear Reviewer,

Thank you for your valuable feedback and the critical points you've highlighted regarding the manuscript. I appreciate the opportunity to refine the document further and address your concerns comprehensively.

Regarding your first point about the use of causal language, I understand the importance of precision in presenting research findings, especially in a study like ours where direct causality cannot be conclusively established. To address this, I have carefully revised the language throughout the manuscript to eliminate words that might imply a direct cause-and-effect relationship, such as "influence", "affect", and "impact". Instead, I've employed terms that more accurately reflect the associative relationships observed in our data. For instance, the sentence in the conclusion now reads:

"This research highlights the associations between fatigue, demographic factors, and the sexual functioning of women with MS, illustrating a complex relationship between physical symptoms and sexual well-being."

For your second point about the formatting of numbers, I have meticulously reviewed the manuscript and corrected inconsistencies in the use of zeroes preceding decimal points and the placement of commas in large numbers. Numbers are now formatted correctly to enhance clarity and ensure adherence to journal guidelines.

Regarding the use of commas and full stops in numbers, adjustments were made to comply with the standard numerical format preferred in the journal's style guide. This should resolve any confusion regarding numerical values and their interpretation.

Finally, concerning the inconsistencies in the use of abbreviations, I have standardized all abbreviations throughout the text. Specifically, the abbreviation "Fatigue Severity Scale (FSS)" is now consistently introduced at its first mention and used uniformly thereafter. I have ensured that all abbreviations are introduced properly and used consistently to prevent any ambiguity.

I hope these revisions meet your expectations and significantly improve the clarity and accuracy of the manuscript. I look forward to your further suggestions or approval to proceed towards publication.

Kind Regards,